# OpenReview forum: "VaPR - Vision-language Preference alignment for Reasoning"
_colmweb.org/COLM/2025/Conference — COLM 2025_

### Official Review · Reviewer_gWgY · 2025-04-26

**Rating:** 6
**Confidence:** 4
**Ethics Flag:** 1

**Summary:**

This paper studies how preference dataset should be created for Large Vision-Language Models (LVLMs) fine-tuning.
The authors introduce a hard-negative response generation framework that produces rejected responses with targeted errors, maintaining stylistic and length similarity to the accepted ones. Using this framework, the authors develop the VaPR dataset, comprising 30K high-quality samples, to fine-tune three LVLM families.. The resulting models achieve performance improvements across ten benchmarks. The authors conduct scaling analysis, which shows that performance consistently improves with data size.

**Questions To Authors:**

[Q1] While the authors discuss the scaling of the proposed method, Figure 5 does not seem to use a logarithmic scale. Also, x-axis is {3k, 10k, 30k}, but this these three values are placed at the same interval. This is strange to me and makes the readers understand how the proposed method scale for training set size. Could you explain why the authors decided to use this Figure 5 and how this relates to so-called scaling law discussed in the existing work?

**Reasons To Accept:**

1. I think it is a great contribution to create a such a preference dataset. I think the authors' idea to generate rejected responses while maintaining stylistic and length similarity to the accepted ones is reasonable and interesting.

2. The performance improvement after applying the dataset is significant. The authors conducted extensive experiment in many benchmark problems using several models. I think the empirical evaluation is reliable.

**Reasons To Reject:**

1. While I do admit that releasing new datasets is a huge contribution, the data generation process in this paper is reasonable yet not very novel.

2. To be honest, I would like to see more examples of the dataset and check the source code. I am not fully sure whether this is possible according to the COLM rule.

### Minor comments
- Please properly use \citep or \citet for citations.

---

> ### Author Response · Authors · 2025-06-03
>
> We thank the reviewer for their insightful feedback, and for recognizing the contribution of our (a) dataset, particularly the novel idea of generating rejected responses with stylistic and length similarity to accepted ones, (b) extensive experiments and reliable empirical evaluation. Below, we address your questions and concerns.
>
> ### Q1. While I do admit that releasing new datasets is a huge contribution, the data generation process in this paper is reasonable yet not very novel.
>
> While several prior works have explored synthetic preference data, our contribution is the dataset and data generation framework, which distinguishes itself in the following key ways:
>
> - **Task-Aware Perturbations**: Unlike prior works (e.g., POVID, SIMA, CSR, RLAIF-V), our VaPR framework generates hard negative rejected responses with task-aware semantic errors, while explicitly preserving stylistic and length consistency. This prevents models from exploiting superficial cues during preference optimization (Section 4.3.1, Table 1, 2 in the main paper).
> - **Response Editing Approach**: While existing approaches rely on VLMs (the same or a large oracle VLM) to generate and/or score sampled responses for preference datasets, VaPR instead leverages large, high-quality open-source SFT datasets to generate targeted negatives by editing ground-truth responses use LLMs. This leverages LLM’s’ semantic understanding of text, which is typically more reliable than image-text comprehension in VLMs [1].
>
> These contributions make our pipeline particularly effective for post-SFT preference alignment and distinguish it from existing approaches. As we demonstrated in response to Reviewers iz1D and W6PJ, our process generalizes to open source models, where models trained on VaPR-OS (generated using Qwen3-32b [2]) achieve ~ 99% of performance of those trained VaPR (generated using GPT-4o). This means that many researchers could adapt our work to their own internal SFT datasets, without relying on closed-source APIs and potentially gain improvements beyond the ones made possible by our other main contribution: the VaPR dataset.
>
> ### Q2. To be honest, I would like to see more examples of the dataset and check the source code. I am not fully sure whether this is possible according to the COLM rule.
>
> We appreciate the reviewer’s interest in seeing more examples of the dataset and accessing the source code. As detailed in Appendix Section B.2, our illustrations show the generation steps for various tasks. For tasks requiring a penalty list - color and counting (to ensure diversity of perturbations) and captioning (to ensure diversity of perturbed dimensions like size, spatial relation, etc), we provide a PDF with additional examples highlighting how the penalty list is applied. We use the color task as an example. Additionally, we include source code for generating samples with and without a penalty list using GPT-4o, with color and spatial relation serving as representative tasks, respectively. Note that the shared scripts assume task-specific samples are pre-filtered (Section 3.2) and handle model input format post-processing separately.
>
> OSF  Link - Contains the PDF and code scripts: https://osf.io/pfz72/?view_only=57f0fe6e0b384bf4a6b165cb69aea753
>
> ### Q3. Could you explain why the authors decided to use this Figure 5 and how this relates to so-called scaling law discussed in the existing work?
>
> We thank the reviewer for raising this point. We acknowledge that Figure 5 does not use a logarithmic scale, and that the x-axis spacing may give the impression of regular intervals despite the non-linear values (3k, 10k, 30k). Our intention was not to claim that our data necessarily follows the same scaling laws as pretraining but rather to illustrate how performance gains vary with increased data size within our budget constraints. We agree that the term “scaling law” may be misleading in this context and will clarify the wording and the figure in the revised paper to avoid confusion. Our goal is to provide initial insights that can inform the community to further explore scaling opportunities if appropriate.
>
> ### Q4. Please properly use \citep or \citet for citations.
> Thank you for pointing this out, we will correct citation formatting to consistently use \citep and \citet in the revised paper.
>
> **References**
> - [1] Guan, Tianrui, et al. "Hallusionbench: an advanced diagnostic suite for entangled language hallucination and visual illusion in large vision-language models." Proceedings of the IEEE/CVF Conference on Computer Vision and Pattern Recognition. 2024.
> - [2] Yang, An, et al. "Qwen3 technical report." arXiv preprint arXiv:2505.09388 (2025).

---

> ### Comment · Reviewer_gWgY · 2025-06-08
> **Acknowlegement**
>
> Thank you for providing the authors' feedback. I have read other reviews and rebuttals. The responses mostly addressed my concerns, so keep my original rating of 6.

---

> > ### Author Response · Authors · 2025-06-08
> >
> > We thank the reviewer for their service and for acknowledging that our rebuttal addressed their concerns. We appreciate their thorough review and will ensure that the revised paper incorporates the additional ablations and recommended clarifications to strengthen it further. If there are any remaining points or additional suggestions, we would be happy to consider them and address them accordingly.

---

### Official Review · Reviewer_W6PJ · 2025-05-11

**Rating:** 7
**Confidence:** 2
**Ethics Flag:** 1

**Summary:**

This paper introduces VAPR, a new dataset and training framework designed to improve preference alignment in Large Vision-Language Models (LVLMs) via hard-negative generation. The authors construct a 30K-sample dataset where the rejected responses are synthetically generated to closely match the length and style of the accepted ones but contain task-specific semantic errors. The resulting models, fine-tuned via DPO across LLaVA and Qwen2VL families, demonstrate consistent improvements across 10 benchmarks, with particular gains in reasoning and adversarial settings.

**Questions To Authors:**

The VAPR SFT results sometimes degrade performance, especially in strong pretrained models (Qwen2.5VL). While DPO seems to mitigate this, it raises questions about how sensitive the models are to overfitting on synthetic negatives.

How does VAPR perform when combined with other reward modeling approaches (e.g., PPO or GRPO)?

**Reasons To Accept:**

The paper clearly articulates a real and pressing issue in preference data construction for LVLMs—namely, that stylistic/length biases contaminate learning. The dataset can have an impact on post-training of vision language models.

VAPR-tuned models show consistent performance gains on vision-centric benchmarks such as SEED, CV-Bench, and NaturalBench with different backbones such as Llava and Qwen-2-VL, which outperform strong baselines including SIMA, CSR, and RLAIF-V.

The paper includes thoughtful metrics like Levenshtein distance, log-probability gaps under the reference model, and reward saturation curves, These results provide additional empirical evidence to verify that VAPR mitigates shortcut learning and improves true preference understanding.

**Reasons To Reject:**

The method depends on GPT-4o’s ability to precisely introduce task-relevant semantic errors. There is no ablation on using weaker or open-source editors, leaving it unclear whether the pipeline is usable in constrained settings.

It would be even better if the authors could provide some additional analysis on why this proposed method will lead to performance improvements. For example, which types of tasks will benefit from the synthetic data?

---

> ### Author Response · Authors · 2025-06-03
>
> We thank the reviewer for their thoughtful and constructive feedback. We appreciate your positive assessment of our work and are pleased that you found the (a) dataset, (b) framework, (c) model comparison and (d) metrics & analysis to be meaningful and impactful. Below, we address your specific concerns.
>
> ### Q1. There is no ablation on using weaker or open-source editors, leaving it unclear whether the pipeline is usable in constrained settings.
>
> We agree that understanding the pipeline’s generalizability beyond GPT-4o is important. To this end,  we have applied our VaPR pipeline using the open-weight Qwen3-32b model [1] - focusing on the following task categories: color, object, spatial, counting, size, background, existence and referential VQA.
>
> *Table 1: Performance is compared across LLaVA-v1.5-Instruct, Qwen2VL-Instruct, Qwen2.5VL-Instruct, and DPO models, all preference finetuned on VaPR (10K subset created with GPT-4o) or VaPR-OS (8K samples generated by the Open-source model Qwen3-32b). Aside from the model used for generation, the framework and prompts are identical. All models share the same hyperparameters. Higher scores indicate better performance, with the top result for each benchmark shown in **bold**. In case two models get the top scores, both are **bolded**, otherwise top score is **bolded** and the second highest score(s) is $\underline{underlined}$*
>
> | Row | Method                   | LLaVA(W) | ConT            | MMV            | SEED(I)           | CV              | MV             | MMMU           | MMS             | POPE            | NB             |
> |-----|--------------------------|---------|----------------|----------------|-------------------|-----------------|----------------|----------------|-----------------|-----------------|----------------|
> | 1   | LLaVA-1.5-7B-Instruct     | 64.8    | 16.8           | 30.9           | 66.2              | 62.1            | 30.1           | 35.4           | 32.6            | **85.9**        | 12.7           |
> | 2   | + VaPR-OS DPO            | $\underline{73.3}$ | $\underline{18.7}$ | $\underline{32.1}$ | $\underline{66.3}$ | **62.3**       | $\underline{30.2}$ | **35.6**       | $\underline{33.7}$ | 83.6          | $\underline{13.9}$ |
> | 3   | + VaPR DPO               | **74.4** | **20.2**       | **32.3**       | **66.4**          | **62.3**        | **30.4**       | **35.6**       | **34.0**        | $\underline{85.2}$ | **14.0**       |
> | 4   | Qwen2VL-2B-Instruct      | 83.2    | 27.7           | $\underline{53.3}$ | 73.6            | 66.5            | **51.0**       | 38.7           | 43.4            | 86.5            | 24.3           |
> | 5   | + VaPR-OS DPO            | $\underline{84.1}$ | $\underline{32.8}$ | $\underline{53.3}$ | $\underline{73.7}$ | $\underline{67.9}$ | 50.2       | $\underline{38.9}$ | **43.5**        | $\underline{88.0}$ | **25.2**      |
> | 6   | + VaPR DPO               | **84.3** | **33.2**       | **53.4**       | **73.8**          | **68.3**        | $\underline{50.5}$ | **39.0**     | **43.5**        | **88.2**        | **25.2**       |
> | 7   | Qwen2.5VL-3B-Instruct    | **98.1** | 37.2           | **67.3**       | 75.0              | 71.5            | $\underline{52.5}$ | **45.7**     | 54.7            | **86.3**        | 25.4           |
> | 8   | + VaPR-OS DPO            | 95.7    | $\underline{39.0}$ | 65.4        | **75.3**          | **72.0**        | 52.4           | $\underline{45.5}$ | $\underline{55.4}$ | $\underline{86.2}$ | **25.7**      |
> | 9   | + VaPR DPO               | $\underline{96.5}$ | **39.3**       | $\underline{66.9}$ | **75.3**          | **72.0**        | **52.7**       | 45.1            | **55.6**        | 86.1            | **25.7**       |
>
>
> We preference finetuned LLaVA-v1.5-Instruct-7B, Qwen2VL-Instruct-2B and Qwen2.5-VL-Instruct-3B on VaPR-OS (VaPR Open source), where the difference between the models are two aspects: (a) is missing reasoning and captioning samples (1K each), reason shared below (b) uses an open-weight model for generating responses. Key findings include:
>
> - **Dataset Quality**: VaPR-OS rejected responses exhibit similar hard-negative properties to those generated by GPT-4o, with an average token length difference of 6 (compared to 3 in VaPR) and a Levenshtein distance of 10 (vs. 6 in VaPR).

---

> > ### Author Response · Authors · 2025-06-03
> >
> > - **Model Performance**:
> >   - Models trained on this open-weight dataset perform comparably to those generated with GPT-4o, where both models independently outperform the baseline on majority of the benchmarks. This is expected, as VaPR-OS samples overlap with VaPR, and dataset statistics show similar linguistic similarity and average token length. This underscores the VaPR pipeline generalizes effectively and is not restricted to closed-weight models.
> >   - Interestingly, we find that the contribution of captioning and abstract reasoning tasks to preference learning is limited, which is consistent with prior work [3] suggesting that complex tasks like reasoning may benefit stepwise decomposition of preference datasets. In our case, captioning and abstract reasoning tasks (e.g. Considering the presence of two clocks on the building, what purpose might this architectural design serve?), can be decomposed into simpler components like fine-grained perception (e.g., attribute recognition) and spatial reasoning (e.g., object location). Training models using these atomic tasks as step-wise preference samples may collectively support learning for more complex tasks, a direction we plan to investigate further.
> > - **Limitations and Future Directions for OS editors**: We observed that data generated using Qwen3 sometimes fails to consistently perturb dependent spans (e.g., object perturbation: “... bathroom … has a large bathtub …” → “... kitchen … has a large bathtub …”, whereas GPT-4o correctly changes it to “... kitchen … has a large oven …”), with this issue being prominent in captioning and abstract reasoning tasks. This could potentially add noise to the dataset and thus we omit these samples in our analysis. To mitigate this issue, we plan to experiment with more open-source models and generation of step-wise preference datasets.
> >
> > These results demonstrate that our pipeline generalizes effectively to different LLM editors and is not restricted to closed-weight models. We plan to add these results and discussions in the revised paper.
> >
> >
> > ### Q2. some additional analysis on why this proposed method will lead to performance improvements. For example, which types of tasks will benefit from the synthetic data?
> >
> > We provide the breakdown of model performance across different task categories for two comprehensive dataset MMStar and CV Bench.
> >
> >
> > *Table 1: MMStar Performance is compared across LLaVA-v1.5-Instruct, Qwen2VL-Instruct, Qwen2.5VL-Instruct, and DPO models – preference fine-tuned on name (30K subset created with GPT-4o). Higher scores indicate better performance, with the top result for each benchmark shown in bold. All models share the same hyperparameters. Abbreviations: CP = Coarse Perception, FGP = Fine-grained Perception, IR = Instance Reasoning, LR = Logical Reasoning, S&T = Science & Technology.*
> >
> > | Model                        | Final Score | CP   | FGP  | IR   | LR   | S&T  | Math |
> > |------------------------------|-------------|------|------|------|------|------|------|
> > | LLaVA-v1.5-Instruct-7B       | 32.6        | 58.8 | 26.8 | 40.0 | 26.0 | 17.2 | 26.8 |
> > | VaPR-LLaVA-7B            | **34.7**    | **62.0** | **27.2** | **44.0** | **27.2** | **18.0** | **30.0** |
> > | LLaVA-v1.5-Instruct-13B      | 33.8        | 58.0 | 27.2 | 42.4 | 26.4 | 21.2 | **27.6** |
> > | VaPR-LLaVA-13B           | **35.6**    | **60.8** | **28.4** | **47.6** | **28.0** | **23.6** | 25.2 |
> > | Qwen2VL-Instruct-2B          | 43.4        | 52.4 | 41.6 | **51.6** | **43.2** | 31.2 | **40.4** |
> > | VaPR-Qwen2VL-2B         | **43.7**    | **53.6** | **45.6** | 50.0 | 42.0 | **31.6** | 39.6 |
> > | Qwen2VL-Instruct-7B          | 56.7        | 67.2 | 50.4 | 62.8 | 56.4 | 46.4 | 57.2 |
> > | VaPR-Qwen2VL-7B          | **57.8**    | **67.6** | **51.2** | **63.2** | **57.6** | **49.2** | **58.0** |
> > | Qwen2.5VL-Instruct-3B        | 54.7        | 66.4 | 46.4 | 60.8 | 54.8 | **39.6** | 60.4 |
> > | VaPR-Qwen2.5VL-3B        | **56.1**    | **68.4** | **47.2** | **63.6** | **55.2** | **39.6** | **62.4** |
> > | Qwen2.5VL-Instruct-7B        | 61.9        | **72.0** | 54.0 | **70.8** | 63.2 | 44.8 | 66.4 |
> > | VaPR-Qwen2.5VL-7B        | **62.5**    | 71.6 | **55.2** | 70.0 | **64.4** | **45.6** | **68.4** |

---

> > > ### Author Response · Authors · 2025-06-03
> > >
> > > *Table 2: CV Bench Performance is compared across LLaVA-v1.5-Instruct, Qwen2VL-Instruct, Qwen2.5VL-Instruct, and DPO models – preference fine-tuned on name (30K subset created with GPT-4o). Higher scores indicate better performance, with the top result for each benchmark shown in bold. All models share the same hyperparameters. Abbreviations: Overall = Overall Accuracy, Count = Count Accuracy, Spatial = Spatial Relation Accuracy, Depth = Depth (Order) Accuracy, Distance = Relative Distance Accuracy. Depth: Determine which of the two distinct objects is closer to the camera, and Relative Distance: Determine which of the two distinct objects is closer to the anchor object.*
> > >
> > > | Model                        | Overall | Count | Spatial | Depth | Distance |
> > > |------------------------------|---------|-------|---------|-------|----------|
> > > | LLaVA-v1.5-Instruct-7B       | 62.2    | 54.3  | **71.2** | 70.0  | **56.3** |
> > > | VaPR-LLaVA-7B            | **62.9**| **57.2** | 70.8  | **71.7** | 54.3     |
> > > | LLaVA-v1.5-Instruct-13B      | 62.5    | 58.8  | 68.2    | 69.7  | 55.8     |
> > > | VaPR-LLaVA-13B           | **64.6**| **58.9** | **69.9** | **72.7** | **59.7** |
> > > | Qwen2VL-Instruct-2B          | 66.5    | 66.6  | 67.5    | 65.8  | 67.5     |
> > > | VaPR-Qwen2VL-2B          | **69.0**| **68.2** | **68.3** | **68.5** | **72.2** |
> > > | Qwen2VL-Instruct-7B          | 75.7    | **67.0** | 79.5  | **85.5** | 72.8     |
> > > | VaPR-Qwen2VL-7B          | **76.3**| 66.6  | **81.9** | 82.7  | **76.3** |
> > > | Qwen2.5VL-Instruct-3B        | 71.5    | 68.5  | 74.8    | 78.2  | 66.3     |
> > > | VaPR-Qwen2.5VL-3B        | **72.7**| **68.9** | **75.1** | **79.3** | **69.2** |
> > > | Qwen2.5VL-Instruct-7B        | 80.1    | 68.5  | 90.0    | 86.7  | 78.5     |
> > > | VaPR-Qwen2.5VL-7B        | **81.1**| **69.2** | **90.6** | **86.8** | **81.2** |
> > >
> > > **Observations**:
> > > - From the analysis above (and Table 2 in the main paper), we observe that VaPR models consistently improve on perception tasks  (particularly fine-grained perception), and reasoning tasks, such as spatial relationships (even complex ones like distance and depth) and counting. These gains align with their improved performance on SeedBench and NaturalBench, both of which emphasize visio-linguistic compositionality, perception, and reasoning.
> > >
> > > - Notably, despite not being explicitly trained on OCR, textual reasoning, or math tasks, VaPR models achieve strong performance in these areas. We attribute this to their enhanced fine-grained perception, spatial reasoning, and counting capabilities, which demonstrates that improvements in these areas support interpretation of embedded text (consistent with prior work [3]) and geometric figures. This trend is further corroborated by gains on ConTextual and MathVista benchmarks. Interestingly, VaPR-Qwen2VL-2B achieves the largest gains on Pope, which can be explained by pronounced improvements in fine-grained perception (as evident in MMStar). On the other hand, it shows slight degradation in math, logical reasoning tasks, which can explain why it does not improve on MathVista, where the other models do (Table 2 in the main paper).
> > >
> > > - Lastly, prior work [3] indicates that preference optimization primarily enhances truthfulness and alignment, rather than factuality, which aligns with our observed improvements in perception and reasoning but not in purely knowledge-based tasks. We hypothesize that limited gains in MMMU (Table 2 in the main paper) can be attributed to the alignment tax VaPR models possibly pay in knowledge based tasks. These results and further discussion will be added to the revised paper and appendix.
> > >
> > > ### Q3. It raises questions about how sensitive the models are to overfitting on synthetic negatives.
> > >
> > > Thank you for the insightful question. Regarding overfitting on synthetic negatives, the hard negatives in VaPR are designed to match chosen responses in length and style (Section 4.3.1 in the main paper), which prevents trivial exploitation of those biases. That said, while we emphasize task diversity in the dataset, we do not yet enforce strict sample-level diversity. Nonetheless, our results show broad improvements across tasks, indicating that the model is generally learning meaningful preference signals rather than overfitting. To further strengthen the dataset and mitigate potential risks, VaPR framework can be extended with an active learning loop (e.g., [4]) that selects the most informative samples, promoting both sample efficiency and diversity. This approach could also be adapted to benefit online RLHF algorithms like PPO.

---

> > > > ### Author Response · Authors · 2025-06-03
> > > >
> > > > ### Q4. How does VAPR perform when combined with other reward modeling approaches (e.g., PPO or GRPO)?
> > > >
> > > > Prior work [3] shows that PPO outperforms DPO in reasoning tasks due to its nature of using chain-of-thought during learning, making it a compelling avenue to explore. GRPO, with its ability to learn from ranked preferences and multiple reward functions, is especially promising for complex, compositional tasks like math [5]. We hypothesize that combining GRPO with VaPR, including expanding VaPR to new tasks and domains, will further enhance performance on benchmarks like MathVista, MMMU, and ConTextual.  We appreciate this suggestion, but due to the time-intensive implementation required for these online algorithms, we plan to explore these extensions in future work.
> > > >
> > > > **References**
> > > >
> > > > - [1] Yang, An, et al. "Qwen3 technical report." arXiv preprint arXiv:2505.09388 (2025).
> > > > - [2] Lai, Xin, et al. "Step-dpo: Step-wise preference optimization for long-chain reasoning of llms." arXiv preprint arXiv:2406.18629 (2024).
> > > > - [3] Fu, Ling, et al. "OCRBench v2: An Improved Benchmark for Evaluating Large Multimodal Models on Visual Text Localization and Reasoning." arXiv preprint arXiv:2501.00321 (2024).
> > > > - [4] Ivison, Hamish, et al. "Unpacking dpo and ppo: Disentangling best practices for learning from preference feedback." Advances in neural information processing systems 37 (2024): 36602-36633.
> > > > - [5] Ji, Kaixuan, Jiafan He, and Quanquan Gu. "Reinforcement learning from human feedback with active queries." arXiv preprint arXiv:2402.09401 (2024).
> > > > - [6] Shao, Zhihong, et al. "Deepseekmath: Pushing the limits of mathematical reasoning in open language models." arXiv preprint arXiv:2402.03300 (2024).

---

> > > > > ### Author Response · Authors · 2025-06-08
> > > > >
> > > > > Thanks again for your insightful feedback on our work! We've carefully worked to address your comments/questions and would like to note that the end of the discussion phase is coming soon. Are there any further questions we should discuss?

---

> > > > > > ### Comment · Reviewer_W6PJ · 2025-06-09
> > > > > >
> > > > > > Thanks for the authors' detailed response. I'll keep my positive score.

---

> > > > > > > ### Author Response · Authors · 2025-06-09
> > > > > > >
> > > > > > > We thank the reviewer for their service and for acknowledging our rebuttal. We will ensure that the revised paper  incorporates the additional ablations, analysis and discussion.

---

### Official Review · Reviewer_iz1D · 2025-05-12

**Rating:** 7
**Confidence:** 4
**Ethics Flag:** 1

**Summary:**

The paper introduces VaPR, a preference dataset of hard negative answers for vision-language tasks generated using a three-step pipeline. The idea is to collect meaningful negative responses as hard negatives while keeping the same length and style as the correct answer. The authors perform several experiments, using this dataset as a training mixture for vision-language models (LLaVA, Qwen VL 2 and 2.5), and test their performance on popular benchmarks.

**Questions To Authors:**

While the paper focuses on stylistic and length differences, it is unclear how this is addressed in the generation step. After checking the Appendix of the work, it seems to me that this is provided as conditioning information, that is, the prompt expressively states to follow guidelines regarding the style and length of the responses. Is this correct? If so, can this be better described when the generation step of the pipeline is showcased?

**Reasons To Accept:**

Models trained on VaPR perform remarkably on popular benchmarks. Additionally, the authors provide comparative results against models trained on alternative datasets.
- A meaningful subset of dataset instances was manually validated by humans, ensuring the high quality of the dataset.
- Extensive study of the generated outputs, with an interesting analysis regarding "Yes/No" questions.

**Reasons To Reject:**

- The proposed pipeline doesn't introduce any meaningful novelty with respect to the state-of-the-art. Many works use Large Language Models for synthetic generation. Additionally, there are works that consider their usage for hard negatives (e.g., in mme5 [1], GPT4-4o was used for plausibility checks of hard negatives obtained from a CLIP model).
The dataset was generated using GPT-4o, limiting the study's reproducibility. Furthermore, the proposed pipeline's generalizability and soundness are limited due to its reliance on GPT-4o. The work would benefit from additional experimentation, potentially incorporating open-weight models.

[1]: Chen, Haonan, et al. "mmE5: Improving Multimodal Multilingual Embeddings via High-quality Synthetic Data." arXiv preprint arXiv:2502.08468 (2025).

---

> ### Author Response · Authors · 2025-06-03
>
> We thank the reviewer for their thoughtful and constructive feedback, as well as for recognizing (a) the strong benchmark performance of VaPR-trained models, (b) our comparative evaluation against alternative datasets, (c) the human validation that ensures dataset quality, and (d) our detailed analysis of generated outputs, particularly for “Yes/No” questions. Below, we address your specific concerns.
>
> ### Q1. The proposed pipeline doesn't introduce any meaningful novelty with respect to the state-of-the-art.
>
> While we acknowledge prior works such as mmE5 that utilize LLMs for generation, our approach focuses on post-SFT data generation and distinguishes itself in the following key ways:
>
> - **Distinction from mmE5**: Unlike mmE5, which generates hard negatives primarily for training upstream embedding models and focuses on plausibility of content, our approach targets post-SFT preference alignment. Importantly, mmE5 does not emphasize length and stylistic consistency, which can be exploited by DPO (Section 4.3.1, Table 1 in the main paper).
>
> - **Distinction from Preference dataset generation methods**:
>    - **Task-Aware Perturbations**: Unlike prior works (e.g., POVID, SIMA, CSR, RLAIF-V), our VaPR framework generates hard negative rejected responses with task-aware semantic errors, while explicitly preserving stylistic and length consistency. This prevents models from exploiting superficial cues during preference optimization (Section 4.3.1, Table 1, 2  in the main paper).
>    - **Response Editing Approach**: While existing approaches rely on VLMs (the same or a large oracle VLM) to generate and/or score sampled responses for preference datasets, VaPR instead leverages large, high-quality open-source SFT datasets to generate targeted negatives by editing ground-truth responses use LLMs. This leverages LLMs’ semantic understanding of text, which is typically more reliable than image-text comprehension in VLMs [1].
>
> These innovations make our pipeline particularly effective for post-SFT preference alignment and set it apart from prior approaches.

---

> > ### Author Response · Authors · 2025-06-03
> >
> > ### Q2. The dataset was generated using GPT-4o, limiting the study's reproducibility.
> >
> > To address your concerns about the pipeline’s reliance on GPT-4o, we have applied our VaPR pipeline using the open-weight Qwen3-32b model [2] - focusing on the following task categories color, object, spatial, counting, size, background, existence and referential VQA.
> >
> > *Table 1: Performance is compared across LLaVA-v1.5-Instruct, Qwen2VL-Instruct, Qwen2.5VL-Instruct, and DPO models, all preference finetuned on VaPR (10K subset created with GPT-4o) or VaPR-OS (8K samples generated by the Open-source model Qwen3-32b). Aside from the model used for generation, the framework and prompts are identical. All models share the same hyperparameters. Higher scores indicate better performance, with the top result for each benchmark shown in **bold**. In case two models get the top scores, both are **bolded**, otherwise top score is **bolded** and the second highest score(s) is $\underline{underlined}$*
> >
> > | Row | Method                   | LLaVA(W) | ConT            | MMV            | SEED(I)           | CV              | MV             | MMMU           | MMS             | POPE            | NB             |
> > |-----|--------------------------|---------|----------------|----------------|-------------------|-----------------|----------------|----------------|-----------------|-----------------|----------------|
> > | 1   | LLaVA-1.5-7B-Instruct     | 64.8    | 16.8           | 30.9           | 66.2              | 62.1            | 30.1           | 35.4           | 32.6            | **85.9**        | 12.7           |
> > | 2   | + VaPR-OS DPO            | $\underline{73.3}$ | $\underline{18.7}$ | $\underline{32.1}$ | $\underline{66.3}$ | **62.3**       | $\underline{30.2}$ | **35.6**       | $\underline{33.7}$ | 83.6          | $\underline{13.9}$ |
> > | 3   | + VaPR DPO               | **74.4** | **20.2**       | **32.3**       | **66.4**          | **62.3**        | **30.4**       | **35.6**       | **34.0**        | $\underline{85.2}$ | **14.0**       |
> > | 4   | Qwen2VL-2B-Instruct      | 83.2    | 27.7           | $\underline{53.3}$ | 73.6            | 66.5            | **51.0**       | 38.7           | 43.4            | 86.5            | 24.3           |
> > | 5   | + VaPR-OS DPO            | $\underline{84.1}$ | $\underline{32.8}$ | $\underline{53.3}$ | $\underline{73.7}$ | $\underline{67.9}$ | 50.2       | $\underline{38.9}$ | **43.5**        | $\underline{88.0}$ | **25.2**      |
> > | 6   | + VaPR DPO               | **84.3** | **33.2**       | **53.4**       | **73.8**          | **68.3**        | $\underline{50.5}$ | **39.0**     | **43.5**        | **88.2**        | **25.2**       |
> > | 7   | Qwen2.5VL-3B-Instruct    | **98.1** | 37.2           | **67.3**       | 75.0              | 71.5            | $\underline{52.5}$ | **45.7**     | 54.7            | **86.3**        | 25.4           |
> > | 8   | + VaPR-OS DPO            | 95.7    | $\underline{39.0}$ | 65.4        | **75.3**          | **72.0**        | 52.4           | $\underline{45.5}$ | $\underline{55.4}$ | $\underline{86.2}$ | **25.7**      |
> > | 9   | + VaPR DPO               | $\underline{96.5}$ | **39.3**       | $\underline{66.9}$ | **75.3**          | **72.0**        | **52.7**       | 45.1            | **55.6**        | 86.1            | **25.7**       |
> >
> > We preference finetuned LLaVA-v1.5-Instruct-7B, Qwen2VL-Instruct-2B and Qwen2.5-VL-Instruct-3B on VaPR-OS (VaPR Open source), where the difference between the models are two aspects: (a) is missing reasoning and captioning samples (1K each), reason shared below (b) uses an open-weight model for generating responses. Key findings include:
> >
> > - **Dataset Quality**: VaPR-OS rejected responses exhibit similar hard-negative properties to those generated by GPT-4o, with an average token length difference of 6 (compared to 3 in VaPR) and a Levenshtein distance of 10 (vs. 6 in VaPR).

---

> > ### Author Response · Authors · 2025-06-03
> >
> > - **Model Performance**:
> >    - Models trained on this open-weight dataset perform comparably to those generated with GPT-4o, where both models independently outperform the baseline on majority of the benchmarks. This is expected, as VaPR-OS samples overlap with VaPR, and dataset statistics show similar linguistic similarity and average token length. This underscores the VaPR pipeline generalizes effectively and is not restricted to closed-weight models.
> >    - Interestingly, we find that the contribution of captioning and abstract reasoning tasks to preference learning is limited, which is consistent with prior work [3] suggesting that complex tasks like reasoning may benefit stepwise decomposition of preference datasets. In our case, captioning and abstract reasoning tasks (e.g. Considering the presence of two clocks on the building, what purpose might this architectural design serve?), can be decomposed into simpler components like fine-grained perception (e.g., attribute recognition) and spatial reasoning (e.g., object location). Training models using these atomic tasks as step-wise preference samples may collectively support learning for more complex tasks, a direction we plan to investigate further.
> > - **Limitations and Future Directions for OS editors**: We observed that data generated using Qwen3 sometimes fails to consistently perturb dependent spans (e.g., object perturbation: “... bathroom … has a large bathtub …” → “... kitchen … has a large bathtub …”, whereas GPT-4o correctly changes it to “... kitchen … has a large oven …”), with this issue being prominent in captioning and abstract reasoning tasks. This could potentially add noise to the dataset and thus we omit these samples in our analysis. To mitigate this issue, we plan to experiment with more open-source models and generation of step-wise preference datasets.
> >
> > These results demonstrate that our pipeline generalizes effectively to different LLM editors and is not restricted to closed-weight models. We plan to add these results and discussions in the revised paper.
> >
> > ### Q3  the prompt expressively states to follow guidelines regarding the style and length of the responses. Is this correct?
> >
> > Thank you for the comment. Yes, this is correct. In our prompts (Appendix Section B.2), we instruct the model to ensure the rejected response is linguistically very similar to the chosen response, with minimal changes that achieve targeted perturbation. This requirement is conveyed implicitly in the prompts, without explicitly mentioning “style” and “length.” Our analysis in Table 1 (main paper) confirms that this strategy effectively maintains style and length similarity, unlike prior works where we see dramatic differences between the chosen and rejected responses, allowing DPO to exploit spurious artifacts (Section 4.3.1 in the main paper). Moreover, our experiments using Qwen3-32b as an open-source editor show that open-source models can follow the same prompting strategy, produce targeted perturbations, and yield models that outperform the base instruct model and achieve ~99% of the performance of the VaPR model (10k subset generated using GPT-4o as the editor)
> >
> > ### Q4 If so, can this be better described when the generation step of the pipeline is showcased?
> >
> > We appreciate the reviewer’s interest in generation examples. As detailed in Appendix Section B.2, our illustrations show the generation steps for various tasks. For tasks requiring a penalty list - color and counting (to ensure diversity of perturbations) and captioning (to ensure diversity of perturbed dimensions like size, spatial relation, etc), we provide a PDF with additional examples highlighting how the penalty list is applied. We use the color task as an example. Additionally, we include source code for generating samples with and without a penalty list using GPT-4o, with color and spatial relation serving as representative tasks, respectively. Note that the shared scripts assume task-specific samples are pre-filtered (Section 3.2 in the main paper) and handle model input format post-processing separately.
> >
> > OSF  Link - Contains the PDF and code scripts: https://osf.io/pfz72/?view_only=57f0fe6e0b384bf4a6b165cb69aea753
> >
> >
> > **References**
> > - [1] Guan, Tianrui, et al. "Hallusionbench: an advanced diagnostic suite for entangled language hallucination and visual illusion in large vision-language models." Proceedings of the IEEE/CVF Conference on Computer Vision and Pattern Recognition. 2024.
> > - [2] Yang, An, et al. "Qwen3 technical report." arXiv preprint arXiv:2505.09388 (2025).
> > - [3] Lai, Xin, et al. "Step-dpo: Step-wise preference optimization for long-chain reasoning of llms." arXiv preprint arXiv:2406.18629 (2024).

---

> > > ### Author Response · Authors · 2025-06-08
> > >
> > > Thanks again for your insightful feedback on our work! We've carefully worked to address your comments/questions and would like to note that the end of the discussion phase is coming soon. Are there any further questions we should discuss?

---

> > > > ### Comment · Reviewer_iz1D · 2025-06-09
> > > >
> > > > Dear authors, thank you for your answers. If the paper is accepted, please clarify the novelty and the difference with respect to existing approaches.

---

> > > > > ### Author Response · Authors · 2025-06-09
> > > > >
> > > > > We thank the reviewer for their service and for acknowledging our rebuttal. We will ensure that the revised paper clarifies the novelty of our data generation framework and its difference with respect to existing approaches. We will also include the additional ablation study and analysis discussed here.

---

### Official Review · Reviewer_2MKa · 2025-05-26

**Rating:** 6
**Confidence:** 4
**Ethics Flag:** 1

**Summary:**

The main contribution of the paper is a creation of balanced preferences dataset for vision-language (VL) tasks. The novelty is creating a preference data for a sample of LLaVA dataset (30K preference pairs). The experiments show performance improvements via DPO optimizaiton compared to multiple baselines. The paper is well written, the contribution is clearly defined, and the empirical investigation is sufficiently thorough. The main concern I have is whether a dataset contribution (moreover, a derived dataset) is a sufficient contribution for a COLM paper.

**Reasons To Accept:**

* Well written paper, and a potentially valuable dataset contribution
* Table 2 demonstrates consistent improvements across benchmarks via DPO
* Nice analysis of things like scaling, and ablations of other approaches, like SFT

**Reasons To Reject:**

* It is not clear whether the contribution of a dataset, while self-contained and significant, is novel enough to warrant an acceptance. Preference-based optimization is known to be effective from prior work, and the dataset construction (sampling + generating a preference pair) is not novel either. **

** The author responses mostly addressed this concern.

---

> ### Author Response · Authors · 2025-06-03
>
> We thank the reviewer for their thoughtful and constructive feedback, as well as for recognizing the value of (a) our dataset, (b) clear presentation (c) thorough empirical analysis. Below, we address your specific concern.
>
>
> ### Q1. It is not clear whether the contribution of a dataset, while self-contained and significant, is novel enough to warrant an acceptance
>
>
> While several prior works have explored synthetic preference data, our contribution is the dataset and **data generation framework**, which distinguishes itself in the following key ways:
>
> - **Task-Aware Perturbations** Unlike prior works (e.g., POVID, SIMA, CSR, RLAIF-V), our VaPR framework generates hard negative rejected responses with task-aware semantic errors while explicitly preserving stylistic and length consistency. This prevents models from exploiting superficial cues during preference optimization (Section 4.3.1, Tables 1 and 2 in the main paper).
>
> - **Response Editing Approach**: While existing approaches rely on VLMs (the same or a large oracle VLM) to generate and/or score sampled responses for preference datasets, VaPR instead leverages large, high-quality open-source SFT datasets to generate targeted negatives by editing ground-truth responses using LLMs. This leverages LLMs' semantic understanding of text, which is typically more reliable than image-text comprehension in VLMs [1].
>
> These contributions make our pipeline particularly effective for post-SFT preference alignment and distinguish it from existing approaches. As we demonstrated in response to Reviewers iz1D and W6PJ, our process generalizes to open source models, where models trained on VaPR-OS (generated using Qwen3-32b [2]) achieve ~ 99% of performance of those trained VaPR (generated using GPT-4o). This means that many researchers could adapt our work to their own internal SFT datasets, without relying on closed-source APIs and potentially gain improvements beyond the ones made possible by our other main contribution: the VaPR dataset.
>
> **References**
> - [1] Guan, Tianrui, et al. "Hallusionbench: an advanced diagnostic suite for entangled language hallucination and visual illusion in large vision-language models." Proceedings of the IEEE/CVF Conference on Computer Vision and Pattern Recognition. 2024.
> - [2] Yang, An, et al. "Qwen3 technical report." arXiv preprint arXiv:2505.09388 (2025).

---

> > ### Comment · Reviewer_2MKa · 2025-06-04
> >
> > Thank you for incorporating these detailed ablations, and detailed explanation of the exact novel contributions. Please make sure to highlight these points on the data generation framework in the paper as well -- it didn't come out as clearly IMO.

---

> > > ### Author Response · Authors · 2025-06-05
> > >
> > > We thank the reviewer for their service and for acknowledging our rebuttal. We appreciate the reviewer’s acknowledgment of our detailed ablations and explanation of the novel contributions. We will ensure that the revised paper highlights these key points about the data generation framework for improved clarity, as recommended.

---

### Decision · Program_Chairs · 2025-07-08

**Decision:**

Accept

**Comment:**

The authors propose a data generation framework via hard negative example synthesis for VLMs. The reviewers all agrees the novelty and the nice analysis in the experiments.